# Vejde: A Framework for Inductive Deep Reinforcement Learning Based on Factor Graph Color Refinement

**Jakob Nyberg**                                                            *jaknyb@kth.se*
*Division of Network and Systems Engineering*
*KTH Royal Institute of Technology*
*Stockholm, Sweden*

**Pontus Johnson**                                                         *pontusj@kth.se*
*Division of Network and Systems Engineering*
*KTH Royal Institute of Technology*
*Stockholm, Sweden*

**Reviewed on OpenReview:** *https://openreview.net/forum?id=EFSZmL1W1Z*

## Abstract

We present and evaluate VEJDE; a framework which combines data abstraction, graph neural networks and reinforcement learning to produce inductive policy functions for decision problems with richly structured states, such as object classes and relations. MDP states are represented as data bases of facts about entities, and VEJDE converts each state to a bipartite graph, which is mapped to latent states through neural message passing. The factored representation of both states and actions allows VEJDE agents to handle problems of varying size and structure. We tested VEJDE agents on eight problem domains defined in RDDL, with ten problem instances each, where policies were trained using both supervised and reinforcement learning. To test policy generalization, we separate problem instances in two sets, one for training and the other solely for testing. Test results on unseen instances for the VEJDE agents were compared to MLP agents trained on each problem instance, as well as the online planning algorithm PROST. Our results show that VEJDE policies in average generalize to the test instances without a significant loss in score. Additionally, the inductive agents received scores on unseen test instances that on average were close to the instance-specific MLP agents.

## 1 Introduction

We are interested in two topics: Deep reinforcement learning (RL) for problem domains that fit relational data models, and agents that can generalize to classes of problems. This interest mainly stems from our experiences in researching automated network incident response, where both of the aforementioned qualities are important for practical use, though not always prioritized (Wolk et al., 2022; Nyberg & Johnson, 2024; Thompson et al., 2024). The two topics are interleaved, in that incorporating structure to the traditional Markov decision process (MDP) formalism typically used in RL is in itself a method for improving agent generalization (Van Otterlo, 2009; Mohan et al., 2024; Kirk et al., 2023), though neither topic necessitates the other. To this end, we have developed a small reinforcement learning library which we have named VEJDE.[1] VEJDE combines graph neural networks and reinforcement learning to incorporate data structures which can be observed or defined in different problem areas, and to produce neural policy functions that can be used for classes of problems defined with a shared data description language.

A decision agent *policy* is a function which takes the state of a decision problem, usually modeled as a MDP or partially observable MDP, as input and produces a probability distribution over actions that are possible

---

[1]Vejde is the Swedish name of the flower *Isatis tinctoria*, traditionally refined to produce indigo dye for textiles.

in the given state. An optimal policy will produce actions that maximize the expected reward; the assigned metric of success at the task the MDP models. Our aim is to find *inductive* policies that can perform well across a domain, or class, of problems. This has long been a goal within the realm of *relational* RL (Dzeroski et al., 2001; Van Otterlo, 2009), which combines reinforcement learning with elements of symbolic logic, but ideas for improving RL generalization have also come from more recent sources in deep learning research (Kirk et al., 2023).

Relational database systems and data modeling remains a popular option for data storage in several areas,[2] which motivates investigating machine learning methods that can utilize the structure relational databases provide. Relational databases use SQL (Chamberlin & Boyce, 1974) and to varying degrees conform with a formal *relational model* (Codd, 1970). Previous work has investigated methods for applying supervised learning to relational databases (Fey et al., 2024b) using graph neural networks (GNNs). From that perspective, this work concerns instead using RL and GNNs with relational databases. Another way of regarding a relational database is as a set of facts about entities described using a typed first-order logical language, where every possible set of facts given the language is a discrete state of some problem domain.

One way of describing a policy for such a logical state representation is as a *decision list*. Each entry in the decision list defines a set of logical conditions, defined in the same language or data model as the database, that when fulfilled leads to an agent choosing an action (Fern et al., 2006). There exists approaches, both symbolic (Fern et al., 2006) and neuro-symbolic (Hazra & Raedt, 2023) which aim to find and represent the decision list in an explicit and human-readable form. We instead opt to use message-passing neural networks (MPNNs) to implicitly encode the rules of a decision list. This choice trades interpretability for flexibility and ease of numerical optimization, as is typical with deep learning methods. The abilities, and limitations, of MPNNs to express logical classifiers has been covered by Barceló et al. (2020) and Grohe (2021).

Thus, in our implementation, the state of the problem consists of a set of known, true, facts about entities expressed in predicate logic. The set of facts is then represented as a bipartite factor graph, and encoded using neural graph color refinement. The color refinement algorithm generates vector embeddings for nodes in the graph through iterative message-passing, where at each step the vector representations of each node is updated by a combination of itself and the graph neighborhood vectors (Morris et al., 2019). A policy head, also using neural networks, predicts the probability of action components from the embedding vectors. The implementation lets VEJDE agents, like a symbolic decision list, handle states with varying numbers of entities and possible actions.

We evaluate VEJDE with eight decision problem domains defined in the Relational Dynamic Influence Diagram Language (RDDL). Each RDDL domain shares a common description language and state transition dynamics, from which ten problem instances with varying parameter values per domain have been defined. The problem domains we include for our evaluation include deterministic and probabilistic state transitions, as well as both discrete and continuous state variables. To test the inductive qualities of the VEJDE policies, we separate problem instances into two sets, where one is used for training and the other for testing. As all problem instances share the same reward and state transition functions, an inductive policy should receive in the same average score on test set as it does on the training set.

We performed two main experiments. One in which we trained VEJDE policies in a supervised manner, using actions provided by another agent as labels, to test if VEJDE can encode an inductive policy for the problem domains. In the second experiment, we trained policies using RL, using the actor-critic algorithm proximal policy optimization (PPO), to test if an inductive policy can be found without labeled data.

We compare VEJDE policies with two other adaptive decision agent types. The first are policies parametrized using fully-connected multi-layer perceptrons (MLPs). Unlike the VEJDE policies, based on GNNs, these take constant-sized and instance-specific vector inputs, and can not generalize across multiple instances. The MLPs encodes policies which are specially developed for a particular problem instance, and which they ideally should be optimal for. Our other point of comparison is the online planning algorithm PROST. PROST represents a different problem-solving methodology than RL, and uses a tree-search algorithm based on

---

[2]According to `https://db-engines.com/en/ranking`.

repeated simulated trials to estimate an optimal action for each encountered state. PROST, for most problems, serves as an upper bound of scores in our evaluation. We also set a lower performance bound for each problem, in the form of random and do-nothing policies, to determine if the learned policies are better than a trivial policy.

The results from the supervised learning experiment showed that VEJDE policies in average generalized to the unseen test set problems without a statistically significant drop in score. The mimic policies are receiver higher scores than the trivial policies on all domains, but the differences to the PROST scores are larger on some problem domains than others. From the reinforcement learning experiment, we observed that the scores of the VEJDE policies in average were not statistically different from those of the MLP policies. However, we also noted that the scores of the RL policies varied a lot between problem domains, with the VEJDE agent not outperforming the trivial policies on two domains. As expected, the RL policies of both types consistently received lower scores than PROST, though on one domain the VEJDE agent received a higher average score than PROST.

Our work builds upon previous work in the area of symbolic, and neuro-symbolic, decision learning, both within the realm of automated planning through machine learning (Ståhlberg et al., 2023; Chen & Thiébaux, 2024) and deep relational reinforcement learning (Janisch, 2024; Sharma et al., 2023). Unlike problems common in symbolic planning research, we focus on problems with uncertain outcomes, formalized as MDPs, and which we assume may include continuous variables.

The source code for VEJDE and the RDDL extension both available publicly[34], to facilitate application to new problem domains.

**Contributions** We present and evaluate the framework VEJDE. The contributions of this work can be summarized as follows:

- A methodology to use deep reinforcement learning with problem areas where relational data is observed or can be defined.

- A neural architecture to encode and parametrize an inductive decision policy for relational data.

- We show that for all the decision problems we evaluate with, the architecture can parametrize an inductive policy and that it can mimic the near-optimal performance of a planning algorithm on some problems.

- We show that for a majority of problems we evaluate with, an inductive policy parametrized using the neural architecture that is better than trivial policies and close to instance-specific policies can be found through reinforcement learning.

- A Python library, named VEJDE, with generic interfaces so that it can be applied to problem domains not defined in RDDL.

## 2 Background

This section covers topics related to knowledge representation, graph neural networks and reinforcement learning relevant to this work. We emphasize that our focus is not on how to design or learn data abstractions for a problem domain, and assume that for a given problem there already exists a data model of object classes, attributes and relations.

### 2.1 Knowledge Representation

*Predicate logic*, or *first-order logic*, has long been used to describe situations and dynamics of decision problems (Reiter, 2001; Russell & Norvig, 2010). Through this text, we will use components of first-order

---

[3]https://github.com/kasanari/vejde
[4]https://github.com/kasanari/vejde-rddl

logic to describe the states of problems. We thus use the following definitions: *Predicates* are used to represent properties or relations. We will denote predicate symbols with capital letters such as $P$ or $Q$. An *atom* is a predicate symbol combined with tuple of *variables*, denoted with lowercase letters. Predicate symbols have an *arity* which defines the length of tuples it can be applied to. We will use the term *object* when referring to instantiations of variables, denoted with lowercase letters and a subscript, like $x_1$ or $y_1$. A *fact* refers to a *ground* atom where all variables have been substituted by objects, as in $P(x_1)$ or $Q(x_1, y_1)$. *Function* symbols map objects or tuples of objects to other elements in the language. Given a function symbol $Z$, we could for instance make the statement $Z(x_1) = 30$. Lastly, we assume a typed logic. This means that each predicate is also associated with a tuple that specifies the *types* of the variables. Each object thus has a type or class. Any ground atom that is the result of combining a predicate symbol with a tuple of objects with mismatched types is defined as false. We will simply use the letter used to identify the object to indicate its type, meaning that $x_1$ and $y_1$ can be read as belonging to different object classes. All the definitions form a *language*, which can be specific to a particular problem domain.

## 2.2 Markov Decision Processes

We choose to represent decision problems as Markov decision processs (Puterman, 1994). A MDP is a formalization of a system where an *agent* makes sequential decisions, under the assumption of potential uncertainty about the decision outcomes. A reward or cost value defines a measure of success at the task the agent should perform. MDPs are assumed to have the Markov property that the state and reward dynamics are dependent only on the immediate previous state and action. The agent is assumed to follow a policy, $\pi$, by which it selects actions given the state it observes. For a finite MDP, an *optimal* policy maximizes the expected discounted return, $G_t = \sum_{i=t}^{T} \gamma^i r_i$. The term *relational* MDP has been used by several authors to describe MDPs where the problem state is structured, factored or explicitly represented using symbolic logic (Guestrin et al., 2003; Diuk et al., 2008; Dzeroski et al., 2001; Van Otterlo, 2009; Kersting & Driessens, 2008). A relational MDP can be described as collection of problems, each individually a MDPs, which all share a common description language.

## 2.3 Reinforcement Learning

To search for an optimal policy, we use PPO; a model-free actor-critic reinforcement learning (RL) algorithm (Schulman et al., 2017). Reinforcement learning (RL) constitutes a class of methods to find, or at least search for, the optimal policy of a MDP (Sutton & Barto, 2018). Different RL approaches mainly alter the methods of selecting actions, what data is used to gather or update the policy and the step intervals between updates. Model-free RL learns policy through selecting actions from the agent policy and updating it solely based the resulting transitions of the MDP. This means that model-free methods requires little to no prior knowledge of the problem dynamics, at the expense of requiring more data than model-based methods, which instead incorporate system knowledge to improve the policy. Value-based RL methods uses data to generate estimates of the state-value $v^\pi(s) = \mathbb{E}_\pi [G_t | S_t = s]$ or action-value $Q^\pi(s, a) = \mathbb{E}_\pi [G_t | S_t = s, A = a]$ functions of the MDP, which the policy is then based on. Policy gradient algorithms instead use data to directly optimize a differentiable function towards the optimal policy. Actor-critic methods combines elements of value estimation with policy gradient methods to improve optimization stability, where an "actor" component produces action probabilities, $\pi(a|s_t)$, and a "critic" component produces a state value estimate, $\tilde{v}(s_t)$. PPO is a policy gradient algorithm that enforces a hard limit on how much the policy can change per update. The actor loss for PPO is calculated as $L_p(s_t, a_t, \theta) = \min (r(\theta) \cdot A(s_t, a_t), \ \text{clip} (r(\theta), 1 - \epsilon, 1 + \epsilon) \cdot A(s_t, a_t))$ where $r(s_t, a_t, \theta) = \frac{\pi_\theta(a_t|s_t)}{\pi'_\theta(a_t|s_t)}$, a ratio between the old and updated policy output which is used to clip the loss. $A$ is the action *advantage*, which may be calculated in different ways, but is typically defined as the difference between the state-value and action-value for the action.

## 2.4 Message-Passing Neural Networks

In order to use policy gradient methods we need a differentiable function that can go from the data contained in the state, which we assume to be relational by design, to action probabilities. We use GNNs for this purpose, specifically MPNNs. The properties of MPNNs are closely related to those of the Weisfehler-Lehman (WL)

algorithm, and we will use that algorithm as the basis for this explanation. The WL algorithm is designed to test if two graphs are isomorphic. The algorithm can also be used to generate graph or node representations for graph similarity and classification tasks. At each iteration, the WL algorithm updates a label of each node to a hash value calculated from the old label and labels from the node neighborhood. This is also known as a graph *color refinement* algorithm (Grohe, 2021), as nodes are assigned metaphorical "colors" which are continuously mixed with neighboring node colors. MPNNs follow the same general color refinement algorithm, but use neural networks rather than hash functions to calculate node "colors", which are represented using high-dimensional vectors. This allows for parameterizing differentiable functions on graphs that can be optimized through gradient descent for common machine learning tasks such as classification or regression. A useful feature of MPNNs is that the model parameters are shared across nodes, meaning that the size of the neural network does not need to change with the size of the input graph. It has been shown that MPNNs can not be more expressive than the WL algorithm (Wang & Zhang, 2022; Morris et al., 2019). As such, if the WL test fails to distinguish two graphs, so will a MPNN. From this result, it follows that a classifier based on MPNNs will always produce the same results if the WL algorithm fails to distinguish two graphs, or subgraphs. This result has been elaborated on in the context of logical classifiers by Barceló et al. (2020) as well as Grohe (2021), showing that a GNN without global a readout are as expressive as a graded first-order classifier.[5] A common categorization in graph learning is *transductive* and *inductive* graph learning. Transductive graph learning aims find a model for a task based on a single graph. If the graph structure changes, or a new graph is introduced, a new model has to be trained. Inductive graph learning instead focuses on solving a task for classes of graphs that share some common underlying data distribution. On a practical level, inductive graph learning typically implies that node representations can not be calculated based on identifiers that do not convey meaning outside the particular graph instance.

## 3 Implementation

VEJDE is formed by four primary components described in this section: Representing the problem state using predicate logic, representing the data base as a bipartite graph, encoding the graph using a graph neural network and lastly using the object encoding vectors to produce action probabilities and a state value estimation. A key implementation detail is that the sizes of both the agent input and output is not fixed, and dependent on the number of facts and entities in the state.

### 3.1 States & Actions

We represent the state of the MDP in a given time step as a set of facts expressed in the domain language[6]. To reduce the size of the state data base, facts that are explicitly known to be false are not included. No information is lost through this choice if we maintain the assumption of full observability in the environment, as there is no need to distinguish between a fact being false or missing from the data base (Reiter, 1977). Actions in the MDP are represented as ground atoms from the same language as the state by assigning a subset of predicates in the domain language as action symbols, $A \subset P$. An example state with actions is shown in Appendix A.

### 3.1.1 Bipartite State Graph

In order to generate object representations through graph color refinement, we first represent the state database as a biparitite graph, which consists of two distinct sets of nodes, $V$ and $U$. $V$ is a set of all facts in the state and $U$ is a set of observed objects. A set of edges $E$ relate facts in $V$ to objects in $U$. Each edge is formed by an object being present in the object tuple of a fact. Following the representation in Chen et al. (2024), each edge between an object $u$ and a literal $v$ is associated with the position of $u$ in the tuple of objects of $v$. A rendering of the state graph representation is shown in Figure 1a, along with possible actions in Figure 1b.

---

[5]"Graded" in this context meaning that a path between two entities needs to exist in the graph.

[6]Since the truthiness and values of literals are assumed to be variable, it would be more correct to denote each fact with a time or step. We do not, however, since our calculations only ever involve the current state.

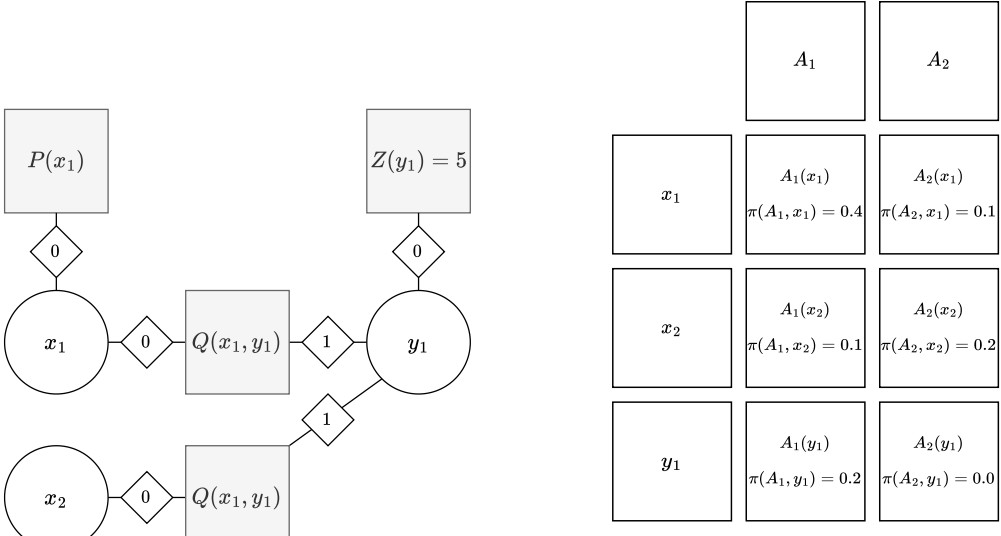

(a) Bipartite graph representation of the set $\{P(x_1),\ Q(x_1, y_1),\ Z(y_1) = 5,\ Q(x_2, y_1),\ C\}$. Squares denote ground expressions, circles objects and rhombuses the position of objects in atoms.

(b) Rendering of action combinations for the state, along with hypothetical action probabilities as assigned by an agent policy.

Figure 1: Renderings of factored state and action representations.

## 3.2 Vector Encoding

All elements of the state are encoded into the vector space $\mathbb{R}^D$ using embedding vectors. The encoded elements form two multisets of initial node embeddings $K = \{\{k_v : v \in V\}\}$ and $H = \{\{h_u : u \in U\}\}$, corresponding to the factor nodes in $V$ and object nodes in $U$. We also encode the positions of objects in the argument list of facts by mapping the positions to vectors in $\mathbb{R}^D$.

### 3.2.1 Literals

Two types of ground expressions are included in our data model; ground atoms, *e.g.* $P(x_1), Q(x_1, y_1)$, and grounded function symbols that map objects to a real value *e.g.* $Z(y_1) = 1$. Ground atoms are encoded by the embedding vector of the predicate symbol. Function atoms are encoded as a linear scaling of the function symbol's embedding vector by the value it is equal to[7]. We can represent the encoding of both types in a single expression by defining the function $\nu : V \to \mathbb{R}$ that maps all elements of $V$ to a value from $\mathbb{R}$. If the element is a ground atom, the value is defined to be 1, and it is an expression with a function symbol it returns the value of the function. Thus, we define the embedding vector $k_v$ for a literal $v$ as $k_v = \nu(v) \cdot E_P(f_{V \to P}(v)),\ E_P : P \to \mathbb{R}^D$.

Expressions with nullary atoms are handled with an additional step. We denote the set of these literals as $G \subseteq V$, and assume that nullary literals influence all objects. The encoding vectors of the nullary literals are aggregated and added to the initial embedding vector of each object. We use the function

$$g(G) = \sum_{v \in G} \mathrm{Softmax}\left([\phi_1(v')]_{v' \in G}\right)_v \cdot \phi_2(v)$$

to aggregate the vectors, where $\phi_1 : \mathbb{R}^D \to \mathbb{R}$ and $\phi_2 : \mathbb{R}^D \to \mathbb{R}^D$ are implemented using neural networks as by Li et al. (2016).

---

[7]As a simplifying delimitation, function symbols are assumed to always map objects to singular values in $\mathbb{R}$.

### 3.2.2 Objects

To avoid using object identifiers as features, objects are mapped to their respective types through a function $f_{U \to T} : U \to T$, where $T$ is the set of types defined in the language. The type identifiers are then mapped to vectors in $\mathbb{R}^D$. The embedding vector $h_u$ for an object $u$ is thus defined as $h_u = E_T(f_{U \to T}(u))$, $E_T : T \to \mathbb{R}^D$.

### 3.2.3 Positions

Edge embeddings are defined as $e_{uv} = E_N(f_{U,V \to N}(u,v))$, $E_N : N \to \mathbb{R}^D$. The function $f_{U,V \to N}$ takes a fact and an object, and maps them to the set of arities for predicates in the language, $N$.

### 3.3 Message Passing

The message-passing scheme we use for color refinement is similar to previous work on neural message passing on factor graphs (Satorras & Welling, 2021). A single iteration of message passing consists of nodes in $K$ sending messages to neighboring nodes in $H$. $H$ is updated and sends messages back to $K$ which is then also updated. Each step of message passing uses a unique set of four neural networks, which we denote with $\phi$. Incoming messages to a node are aggregated by taking the element-wise maximum over the node neighborhood, $\max_{x \in N(y)} : D \to D$. While a max aggregation is theoretically less expressive than a sum aggregation (Xu et al., 2019), it has been found to work well in practice in previous work (Janisch, 2024; Ståhlberg et al., 2023). Messages from $U$ to $V$ are defined as $m_{v \to u}^{(i+1)} = \phi_{V \to U}^{(i)}(h_u^{(i)} \parallel k_v^{(i)})$, and the reverse as $m_{u \to v}^{(i+1)} = \phi_{U \to V}^{(i)}(k_v^{(i)} \parallel h_u^{(i+1)} \parallel e_{uv})$. Representation vectors for object nodes in $U$ are calculated at each iteration as

$$H^{(i+1)} = \left\{\!\!\left\{ h_u^{(i+1)} = \phi_U^{(i)}\left( h_u^{(i)} \parallel \max_{v \in N(u)}(m_{v \to u}^{(i+1)}) \right) : u \in U \right\}\!\!\right\}$$

where $\parallel$ to denotes vector concatenation and $\{\!\{\}\!\}$ a multiset. Updates to factor nodes in $V$ are defined as

$$K^{(i+1)} = \left\{\!\!\left\{ k_v^{(i+1)} = k_v^{(i)} + \phi_V^{(i)}\left( k_v^{(i)} \parallel \max_{u \in N(v)}(m_{u \to v}^{(i+1)}) \right) : v \in V \right\}\!\!\right\}.$$

### 3.4 Policy

A policy function can in a more general view be regarded as classifier, which takes a state representation and assigns a probability to each action that is possible given the state. In this setting, the state is represented as a set of facts, which are encoded using embedding vectors and message passing to a *latent* state $H$ of object embeddings. Thus, from the multiset $H$, the policy function should assign a probability to each action that is possible given the objects in $U$ and the action symbols $A$, as described in Section 3.1. Since we limit action symbols to be nullary or unary, all actions contain a predicate symbol, $a \in A \subset P$, and a single object, $u \in U$. Nullary actions have no arguments by definition, but for ease of implementation we define a null object, $\varnothing$, which is the only selectable object for nullary actions. We denote the joint probability over action symbols and entities as $\pi(a, u | H)$, such that $\pi(a_1, u_1 | H)$ would be the probability of the action $a_1(u_1)$ given the set of object embeddings, $H$. The joint action probability can be predicated in full, or factorized in three different ways, which lead to different agent properties (Nyberg & Johnson, 2024).[8] For this work, we solely evaluate the factorization $\pi(a, u | H) = \pi_A(a | H) \cdot \pi_U(u | a, H)$, which corresponds to first sampling a predicate symbol, followed by an object. The conditional of $H$ will be omitted from here onward, for notation brevity. We calculate the action symbol probability, $\pi_A(a)$, as a weighted sum of conditional action probabilities over objects

$$\pi_A(a) := \sum_u^U \pi_A(a | u) \cdot \pi_U(u)$$

---

[8]The main property that changes is the dependency on global readouts. Sampling an object first and then an action symbol based on that choice can be done without considering the entire graph.

where $\pi_A(a|u) := \text{Softmax}^A(W \cdot h_u, \ a)$, $W \in \mathbb{R}^{|A| \times D}$ and $\pi_U(u) := \text{Softmax}^U(W \cdot H, u)$, $W \in \mathbb{R}^D$.

The probability of selecting an object given an action symbol, $\pi_U(u|a)$, is calculated as $\pi_U(u|a) :=$ $\text{Softmax}^U(H \cdot W_a, u)$, $W \in \mathbb{R}^{D \times |A|}$ where $W_a$ is the column vector in $W$ identified by $a$. We use $\text{Softmax}^D(x, i) = e^{x_i} / \sum_j^D e^{x_j}$ to denote the $i$-th element of a vector $x$ where the softmax function is applied over dimension $D$.

### 3.5 Value Estimate

As a part of the actor-critic framework, the architecture should produce an estimate, $\tilde{v}^\pi$, of the state value. We calculate $\tilde{v}^\pi$ as an expectation over the action probabilities and an action-value estimate,

$$\tilde{v}^\pi = \sum_{a \in A} \pi_A(a) \sum_{u \in U} \pi_U(u|a) \, \widetilde{Q}(u, a)$$

where $\widetilde{Q}(u, a) = h_u \cdot W_a$; $W \in \mathbb{R}^{D \times |A|}$, $h_u \in H$. This formulation can be regarded as dividing a single step in the MDP into two sub-steps, where the first consists of the agent choosing a predicate symbol. If we were to extend the number of object arguments for actions, the number of sub-steps would increase to account for the additional choices.

## 4 Evaluation

We evaluate VEJDE using a set of problems defined with the Relational Dynamic Influence Diagram Language (RDDL) and executed in the Python library `PyRDDLGym`[9] (Taitler et al., 2024). RDDL is a description language that can be used for specifying Markov decision problems as dynamic Bayesian networks (Sanner, 2010). We test the ability of VEJDE to model inductive decision policies in two ways: supervised learning from examples and reinforcement learning.

We compare VEJDE policies with policies parametrized using MLPs, and a planning algorithm named PROST (Keller & Helmert, 2013), the winner of the probabilistic track of the 2014 International Probabilistic Planning Competition (IPPC) (Vallati et al., 2015). PROST uses a tree-search algorithm framework to find an optimal action for each state based on a series of simulations initiated from the state. As such, for each state PROST runs thousands of trials to construct the search-tree. We think it should be noted that this is a fundamentally different problem-solving approach than the reinforcement learning method used to train the VEJDE and MLP policy functions. With RL, a simulator is only used during training to find the parameters of the policies, whereas PROST always needs access to a simulator of the problem in order to perform searches.

We use an existing interface for PROST to `PyRDDLGym` for all experiments.[10] For the MLP policies, we used the implementation included in the `pyRDDLGym` set of libraries, which in turn is based on Stable Baselines 3.[11] In the MLP implementation, the state is represented as a vector over all possible facts, the length of which is dependent on the language and number of objects in the problem instance. This means that the total parameter count of the MLPs input and output layers varies between instances and domains, but all use two hidden layers with 64 latent parameters each. The MLP policies are *transductive*, and can not be used for multiple problem instances due to the varying input and output shapes. We thus train one MLP agent for each domain and instance combination.

Including an inductive agent from previous works would have been useful, but we were unable to find a single method that cover the set of problem domains as we are evaluating on. We should note, however, that there is overlap with previous works on subsets of the domains we include (Janisch, 2024; Garg et al., 2019; Sharma et al., 2023). We did evaluate an alternative encoding method, however, based on *graph attention*. The main difference between this approach and the message passing method described in Section 3.3 is that the graph attention method incorporates information from the entire graph in a single step of message passing. We ultimately found that the graph attention method did not perform significantly better than the message

---

[9]`https://github.com/pyrddlgym-project/pyRDDLGym/commit/f7dd1dd`
[10]`https://github.com/pyrddlgym-project/pyRDDLGym-prost/commit/248d5d2`
[11]`https://github.com/pyrddlgym-project/pyRDDLGym-rl/commit/9714392`

passing method, at a significantly higher computational cost. An extended description of the graph attention method, together with evaluation results, is shown in Appendix H.

## 4.1 Problem Selection

RDDL defines problems as a combination of a *domain* description file and an *instance* description file. The domain file primarily defines state transition dynamics and object classes, whereas instance files declares object instances, ground values and parameters to instantiate problems in the domain. VEJDE can be used with the different domains with no modifications other than adjusting the input and output sizes based on each domain description, meaning that we train one agent per domain. RDDL is similar to the Planning Domain Definition Language (PDDL) in that it is a formal description language used to describe decision problems, but adds modeling features such as probabilities, continuous state variables and partial observability.

`PyRDDLGym` hosts a repository of RDDL problem definitions[12]. We evaluated VEJDE on the following problem domains from previous IPPCs: *Elevators (2014)*, *SysAdmin*, *Navigation*, *Traffic (2014)*, *SkillTeaching (2014)*, *AcademicAdvising (2014)*, *CrossingTraffic (2014)* and *Tamarisk*[13]. Each problem domain has ten instances which vary initial values and the number of entities. All problems have a time horizon of 40 steps, after which the MDP is terminated, and all problems have the option for the agent to do nothing. We selected domains based on a set of exclusionary criteria that filter out problems that define continuous actions, action literals with an arity greater than one, state-dependent conditions on actions and lastly problems that implement partial observability. For example, the IPPC 2014 problem *Wildfire* were excluded since it contains the action fluent $put\_out(x, y)$, which has arity of two. Modifying domains specifications to include more problems is possible, but we considered it out of scope for this work. While we did not modify the domains, we edited all instance specifications that allow for concurrent agent actions to only permit one action per timestep. This includes subsets of instances in *SkillTeaching*, *AcademicAdvising*, *Elevators* and all instances in *Traffic*. Removing concurrent actions allows us to compare the GNN policy against the *pyRDDLGym* MLP policy on all instances, at the expense of slightly changing the problem premises.[14] The conditions used to select problems are discussed further in Appendix D.

The included domains cover two general types of decision problems: *SysAdmin*, *Elevators*, *SkillTeaching*, *Tamarisk* and *Traffic* are reward maximization, or cost minimization, tasks. *CrossingTraffic*, *Navigation* and *AcademicAdvising* are goal-oriented, where the agent actions should lead to a given condition being fulfilled. Based on the arities of the action literals defined for the problems, we can also separate the eight problem domains into three categories. Ignoring the "do nothing" action, *SysAdmin*, *Traffic* and *AcademicAdvising* have only one unary action and no nullary actions. *Elevators*, *SkillTeaching* and *Tamarisk* have multiple unary actions and no nullary actions. *CrossingTraffic*, *Navigation* have four nullary actions that represent cardinal directions for the agent to move, but no unary actions. The maximum predicate arity in all domains is two. This fact was not part of our inclusion criteria, but came as a consequence of the selected problems.

## 4.2 Data Generation

A RDDL problem instance defines a constant number of objects. As we are interested in use-cases where the total number of objects vary, or is unknown, we sample transitions evenly from multiple problem instances as if they constitute a single relational MDP. To test that the policy generalizes, we divide the 10 problem instances of each domain into two sets by random selection and use one for training and the other for testing, as per common practice in supervised learning. The object counts are not evenly distributed, so to reduce bias we sample five train/test splits and repeat experiments for each split, and calculate the final results as an average over them. For the other agents, PROST and the MLPs, considering a "test" set of problems is not possible, as PROST always performs searches and each MLP agent is tied to a problem instance.

---

[12]https://github.com/pyrddlgym-project/rddlrepository/commit/1a2d3b5
[13]The parenthesized year denotes the version used where multiple are available.
[14]Without concurrent actions for *Elevators*, an agent has to prioritize moving one of two elevators at each time step rather than controlling both simultaneously.

### 4.3 Experiments

To evaluate VEJDE, we performed two main experiments. One experiment in which agents are trained by examples generated from PROST, and one where agents are trained without examples through reinforcement learning. We then compare the scores of VEJDE agents against baseline policies, and other approximate policy methods. Both experiments use the same splits of training and test problems.

#### 4.3.1 Imitation Learning

We tested the capability of VEJDE to encode a near-optimal policy by having agents mimic the actions of the planning algorithm PROST on the training set of problems. The resulting mimic policy was then scored on the test set of problems. To collect training data for each domain, we ran PROST on every instance in the training set for ten episodes, each consisting of 40 state-action transitions with the recommended "IPC2014" configuration. This resulted in 2000 samples for each domain, each of which consists of a state and an action taken by PROST. The mimic agent was then trained through *imitation learning*, where the policy is optimized to simply maximize the probability of actions selected by PROST. This procedure was repeated five times, once for each train/test split. The results of this experiment can be found in Section 5.1.

#### 4.3.2 Reinforcement Learning

As we do not wish to assume access to labeled data or an expert policy, the primary interest for us with this work is to find decision policies using RL. While we solely used PPO for RL, other policy gradient algorithms could be used in theory. In training the agent, we maximize the loss $L = -L_a - c_c L_c + c_h H(\pi)$ through stochastic gradient ascent, where $L_a$ is the actor loss, $L_c$ the critic prediction loss $(\tilde{v} - R_t)^2$ and $H(\pi)$ the entropy of the action distribution given the state. The entropy term serves as a regularization parameter and encourages exploration. Each VEJDE policy was trained with a total of $1\,500\,000$ samples from problem instances in the training set. All MLP policies were trained with $400\,000$ samples per instance with the same hyperparameters for all problems. Though both the MLP and VEJDE agents are trained as stochastic policies, we evaluated them deterministically by picking the most probable action from the action distribution for a given state, rather than sampling the distribution. This choice was made to lower the variance of the results, but it ignores that the policy may assign nearly equal probabilities to some actions. The results of this experiment can be found in Section 5.2.

## 5 Results

Our primary evaluation metric is the average return for agents on problem instances in the test set, which VEJDE agents were not trained on. For transductive agents, *i.e.* PROST and the MLP agents, we only count scores on problems that are in the test set, averaged over the respective train/test splits.

Each RDDL domain uses rewards with different scales and magnitudes. We thus present returns normalized to the range $[0, 1]$, calculated using the method from IPPC 2014 (Vallati et al., 2015). Raw return values for each domain, instance and agent can be obtained from our GitHub repository[15]. All experiments were done on a machine equipped with 32 GB of RAM, an Intel Xeon Silver CPU with 24 cores and an NVIDIA Quadro RTX 4000 GPU.

### 5.1 Imitation Learning

Running PROST on the 90 problems for ten episodes each to collect training data took approximately 6 hours. We then ran PROST for 100 episodes on all 90 problems for scores to use in evaluation, which took roughly two days. Each mimic policy was trained for 1000 epochs over the training data, which was fed to the agent in single batches of 2000 samples. Training mimic policies for all domains took approximately 30 minutes, repeated five times with different train-test splits for a total of 2.5 hours. The mimic policy was then tested on all problem instances for 100 episodes. To summarize the differences between the agents, we performed a

---

[15]https://github.com/kasanari/vejde-rddl-eval

Table 1: Differences in average normalized scores for imitation learning policies. Comparing PROST scores, VEJDE scores on test instances, and scores on train instances. $p_{null}$ is probability according to null distribution produced by permutation testing that $P(|X| \geq |\mu_1 - \mu_2|)$.

| Comparison | $\mu_1 - \mu_2$ | $p_{null}$ |
|---|---|---|
| Test  - Train | 0.01 | 0.82 |
| Test  - PROST | -0.26 | 0.00 |
| Train - PROST | -0.28 | 0.00 |

Table 2: Average normalized score per domain on test instances for PROST and VEJDE policies trained with imitation learning.

| Domain | VEJDE $\mu \pm \sigma$ | PROST $\mu \pm \sigma$ |
|---|---|---|
| SysAdmin | $0.99 \pm 0.03$ | $0.87 \pm 0.12$ |
| SkillTeaching | $0.91 \pm 0.22$ | $0.98 \pm 0.02$ |
| Tamarisk | $0.88 \pm 0.06$ | $1.00 \pm 0.01$ |
| Elevators | $0.87 \pm 0.13$ | $0.99 \pm 0.02$ |
| Traffic | $0.73 \pm 0.16$ | $1.00 \pm 0.00$ |
| CrossingTraffic | $0.63 \pm 0.31$ | $1.00 \pm 0.00$ |
| Navigation | $0.33 \pm 0.48$ | $0.99 \pm 0.03$ |
| AcademicAdvising | $0.21 \pm 0.42$ | $0.75 \pm 0.44$ |

permutation test to compare average mimic scores on train and test instances, as well as PROST scores.[16] The test calculates a distribution of a selected metric, in our case the difference in means, under the hypothesis that the compared sample distributions are equal. It does so by pooling all the samples and calculating the metric for permutations of the samples for a given number of repetitions. If the observed metric is unlikely given the resulting distribution of metrics, then the null hypothesis of the samples have equal means can be rejected. We observed a difference of 0.01 between the agent scores on test problems compared to train problems, which according to the null distribution has a probability of occurring by 83%. We thus have more confidence in the means being equal than not according to the test. The differences between PROST and both the training and test sets are significantly larger, with a difference of -0.26 between PROST and the VEJDE scores on the test set. The differences in mean with the $p$-values from the permutation test is shown in Table 1, and a plot of the null distribution from the test is shown in Appendix F.

From the average test scores per domain, we observed varying differences to PROST between problem domains. On some domains, such as *SkillTeaching* and *SysAdmin* there was not a significant difference between the scores of PROST and the mimic policy. Two VEJDE scores fall under and average of 0.5, those for *AcademicAdvising* and *Navigation*. We noted that PROST barely performs better than the lower bound return on certain instances of *AcademicAdvising*, which is a possible cause of the low mimic performance on this domain. Mimic test scores for each domain are shown in Table 2.

## 5.2   Reinforcement Learning

Training VEJDE policies for all domains using reinforcement learning took approximately 5 hours, which repeated five times took a total of 35 hours. Training one MLP policy for each of the 90 problem instances took approximately 24 hours. After training, all policies were executed on each of the problem instances for 100 episodes, and the average return per instance was recorded. We used the same evaluation data for PROST as was used in the imitation learning experiment. To create a summary score for over all domains, we performed a permutation test with $10^6$ samples to compare the test problem scores of the VEJDE agents with the scores of the MLP policies and PROST on the same problems, with the null hypothesis that the agents have equal average scores. If we assume that we can reject the null hypothesis with $p < 0.05$, we can not do

---

[16]We used `https://docs.scipy.org/doc/scipy/reference/generated/scipy.stats.permutation_test.html` to run the test.

Table 3: Difference in average score between VEJDE RL agents, instance-specific MLP agents and PROST on test instances, averaged across all problem domains. $p_{null}$ is probability according to null distribution produced by permutation testing that $P(|X| \geq |\mu_1 - \mu_2|)$.

| Comparison | $\mu_1 - \mu_2$ | $p_{null}$ |
|---|---|---|
| VEJDE - MLP | 0.05 | 0.17 |
| VEJDE - PROST | -0.43 | 0.00 |
| MLP - PROST | -0.48 | 0.00 |

Table 4: Average normalized scores on test instances for agents trained using RL and PROST, per problem domain.

| Domain | VEJDE $\mu \pm \sigma$ | MLP $\mu \pm \sigma$ | PROST $\mu \pm \sigma$ |
|---|---|---|---|
| SkillTeaching | $0.84 \pm 0.10$ | $0.83 \pm 0.12$ | $0.99 \pm 0.02$ |
| AcademicAdvising | $0.77 \pm 0.39$ | $0.19 \pm 0.39$ | $0.41 \pm 0.46$ |
| SysAdmin | $0.75 \pm 0.23$ | $0.89 \pm 0.11$ | $0.94 \pm 0.09$ |
| Traffic | $0.61 \pm 0.24$ | $0.11 \pm 0.12$ | $1.00 \pm 0.00$ |
| Tamarisk | $0.56 \pm 0.26$ | $0.61 \pm 0.11$ | $1.00 \pm 0.00$ |
| CrossingTraffic | $0.42 \pm 0.19$ | $0.45 \pm 0.25$ | $1.00 \pm 0.00$ |
| Navigation | $0.01 \pm 0.03$ | $0.13 \pm 0.33$ | $1.00 \pm 0.00$ |
| Elevators | $0.00 \pm 0.00$ | $0.21 \pm 0.19$ | $1.00 \pm 0.00$ |

so for the difference between the mean scores of the MLP and VEJDE agents, but we can for the difference between PROST and the other agents. There was thus, in average, not a significant statistical difference between the average scores of the VEJDE agents and the MLP agents, which we consider good given that the VEJDE agents were not trained on the considered instances. Both RL agents have significantly worse scores than PROST in average. The statistics of the test are shown in Table 3, and we have included a plot of the resulting null distribution from running the test in Appendix F.

We observed the VEJDE agent achieved significantly higher scores than the MLP agents on the domain *Traffic*, and on one domain, *AcademicAdvising*, the VEJDE agent had higher scores than both PROST and the MLPs. On the domain *Navigation* and *Elevators* the VEJDE agents had significantly worse scores than the other agents, as well as their imitation learning counterparts. The scores of the MLP agents on these domains, though slightly higher than those of the VEJDE agents, are also relatively low, indicating a general difficulty in finding policies for these domains using reinforcement learning. Normalized test scores for each domain are shown in Table 4.

# 6 Related Work

There have been a number of previous works that combine machine learning on graphs with decision problems, and VEJDE piece-wise overlaps with several of these. We summarize the primary differences and overlapping features with a non-exhaustive selection of related works in Table 5.

There exists a large body of work in the realm of *relational reinforcement learning*. The book by Van Otterlo (2009) covers much of research in the area prior to the 2010s, which is composed of both purely symbolic and non-neural machine learning methods for both value-based and policy-based relational reinforcement learning. The survey by Mohan et al. (2024) covers more recent works, as well as some older, from a more general perspective of incorporating structure in RL. They use the term "side information" to refer to various forms of structural information provided to agents to improve learning, but which is not part of the typical MDP formalism. Side information used by various works include data abstractions, information about system dynamics and goal formulations, which typically improve data efficiency or policy generalization at the cost of requiring more prior knowledge about the problem or computational power. By their categorization, the only side information we use in VEJDE is data abstraction, where we assume that there exists a data description language that can describe object classes and relations from a given problem domain. This is, in our opinion, a significantly lighter assumption than for instance Sharma et al. (2023) makes, which is full knowledge of the dynamic Bayesian network in order to construct the input graph for the policy.

An alternate method of inductive policy generation which we consider interesting is decision list learning. The state is represented with symbolic logic, but rules for action selection are represented explicitly in a human-readable manner rather than implicitly encoded as weights in a neural network. Fern et al. (2006) presents a symbolic method that searches for a policy through a combination of beam-searching and policy iteration. Neural methods for decision list rule generation that incorporate numeric optimization have also been proposed (Hazra & Raedt, 2023; Delfosse et al., 2023), which in some cases assign numeric weights to the rules in order to make the policy probabilistic. Representing rules in a human-readable manner has an immediate benefit of being able to manually analyze and interpret the policy, which is good for transparency, but requires a method for searching a potentially large space of possible rules. We also recognize the possibility to extract explicit rules from a trained MPNN policy *post-hoc.*

The topic of generalization in reinforcement learning for classes of problems has also been covered from the perspective of deep learning. In their survey of RL generalization, Kirk et al. (2023) separate learning problems into three categories: *singleton* problems where the training and test environments are the same; Independent and Identically Distributed (IID) generalization problems, where the test environments are different but sampled from the same distribution as the train environments; Out of Distribution (OOD) generalization problems, where the test environments are sampled from a different distribution. An example of such problems is the collection "ProcGen", which define problems using distributions over variables such that randomly generated problem instances can be sampled from them (Cobbe et al., 2020). By their classification, we are focusing on IID generalization of the agent in our evaluation.

## 6.1 Deep Relational Reinforcement Learning

The works which we consider closest in type to ours are those which use graph-based state representations to predict action probabilities in a model-free learning context. The sequential method we use to sample actions is influenced by Janisch (2024), who presents an autoregressive decoding scheme to sample the components of actions, which allows for both nullary and higher-order actions. For actions with an arity greater than one, additional rounds of message-passing are executed after sampling an object, marking objects that have already been selected with an additional feature. Unlike us, they use simple graphs to represent the facts of the state, which limits the method to problems with binary relations. Agents are trained using policy gradient optimization in an unsupervised manner. They evaluate their method on implementations of *SysAdmin*, *Blockworld* and a modified version of the game *Sokoban.*

Garg et al. (2019), like us, use reinforcement learning to train policies based on the RDDL domains *SysAdmin*, *Game of Life* and *Academic Advising.* These are all represented as a simple graph of binary relations between objects, with a single unary action. While this work use a similar evaluation as us, we can not compare our

results directly with them, as they too present normalized scores but have not calculated them in relation to any baseline policies. Therefore, we can not determine if their method performs better than a random or null policy on these domains. Additionally, they train and test on hand-picked subsets of instances, rather than all instances.

Ammanabrolu & Riedl (2019) offers a somewhat different perspective, coming from a natural language processing context. Agents are evaluated on a text-based game, where an agent has to navigate a simulated space and solve puzzles. They represent observations using a simple graph, which use word embeddings as node features, and select actions as Resource Description Framework (RDF) triplets, on the form (subject, verb, object).

We also wish to highlight a set of works we refer to as *object-oriented* reinforcement learning, which includes works by Guestrin et al. (2003), Diuk et al. (2008) and more recently Zambaldi et al. (2019). We categorize these works by the fact that they model the state using conditionally independent objects, but without explicit relationships. Mohan et al. (2024) classifies these works as *factored* representations, as opposed to *relational* representations that include relations, as we do. A number of works in the context of multi-agent reinforcement learning, such as Foerster et al. (2018), arguably fit this category when the complete system state is factorized into discrete states or observations for each individual agent. The object-oriented approach is practical in the sense that one does not have to define or observe relationships between objects, but tends to have worse scaling properties since approaches typically compares every entity to one another. It is, however, less sensitive to violations of the homophily assumptions which MPNNs rely on, *i.e.* that things which are connected in the graph are related.

## 6.2 Automated Planning

Many works that use symbolic logic to represent problems operate in the research area of planning, which tends to assume deterministic system models, as opposed to MDPs with probabilistic state transitions. Nevertheless, we share features with multiple works within this research area. For instance, Chen & Thiébaux (2024) and Ståhlberg et al. (2023) both use factor graph state representations, but predict heuristic scores over subsequent states rather than probabilities over actions. Chen et al. (2024) use the WL algorithm to generate latent states, which are then used to produce heuristic scores. In a subsequent work, Chen & Thiébaux (2024) extends the WL algorithm to incorporate numeric features, and evaluate their method on a single deterministic goal-oriented problem, *Blockworld*. They train their models in a supervised manner based on pre-calculated optimal plans. We find this pair of works interesting in that they show that a computationally simpler representation can be used to generate latent states, at least for the singular problem they investigated. In Ståhlberg et al. (2023) agents are trained in an unsupervised manner using reinforcement learning, as well as a value function using supervised learning. They evaluate their method on several problems defined in PDDL. In a subsequent work, Ståhlberg et al. (2025) focuses on methods for higher-order action selection, but these are only evaluated with supervised learning.

Table 5: Summary of features in a selection of related works in graph-based decision policy learning compared to VEJDE. Only works that base their states on observations, rather than environment dynamics, are included. The "KG" column tells if the state is bounded to only include binary relations, and the "action arity" column shows the number of arguments of actions, when actions are used rather than heuristics.

| Work | Supervised learning | Probabilistic problems | Continous features | KG | Action arity |
|---|---|---|---|---|---|
| Ammanabrolu & Riedl (2019) | No | No | No | Yes | 2 |
| Garg et al. (2019) | No | Yes | No | Yes | 1 |
| Ståhlberg et al. (2023) | No | No | No | No | N/A |
| Chen & Thiébaux (2024) | Yes | No | Yes | No | N/A |
| Janisch (2024) | No | Yes | No | Yes | [0, 1, 2] |
| Ståhlberg et al. (2025) | Yes | No | No | No | 2 |
| Vejde (Our work) | No | Yes | Yes | No | [0, 1] |

# 7 Discussion

This section covers our thoughts on the results, the design decisions made with VEJDE and the problem selection.

## 7.1 Agent Performance

### 7.1.1 Vejde

The VEJDE agent trained with RL received average scores higher than the trivial policies on $8/10$ domains. We consider this a positive indication that VEJDE can be used to represent policies that can generalize to unseen problem instances within a class of problems. We observed a negative difference in score between the mimic policy and RL policy for most problem domains. *Elevators* has the most significant difference, with an 87 percent unit drop between the experiments. Since the problems and VEJDE architectures were the same for both experiments, we attribute this difference to PPO not discovering a set of policy parameters that work and those found when imitating PROST. It remains an open question as to how this difference can be reduced. *Navigation* and *CrossingTraffic* occur in the set of problems the VEJDE policies receive the lowest scores in, both in the reinforcement and imitation learning experiments. They are both grid-based, with discrete cartesian coordinates represented as objects and only nullary actions. These are arguably not the kinds of problems that we would choose to use VEJDE with, in that representing a uniform grid as a graph is needlessly complicated and the number of possible actions is constant regardless of the grid size. Nevertheless, they fit the selection criteria we set, so we feel that it would be unfairly selective to exclude them. They are arguably a good test of the architectures ability to generate whole-graph embeddings, in that they can be regarded as graph classification tasks, but the hypothesis that a whole graph aggregation is needed to solve these problems remains untested.

### 7.1.2 MLP

The transductive MLP policies received higher average scores than VEJDE agents for individual instances, meaning that they represent a good alternative to VEJDE if generalization is not a priority. If the number of entities in the problem changes, additional policies need to be trained to account for the different input space,[17] or an input space that is agnostic to the number of entities need to be used. In theory, a VEJDE policy can only be as good as a MLP policy on a given instance, assuming that the MLP policy is optimal, and the inductive policy is also optimal for all problem instances. If the inductive policy can not be optimal for all instances due to problem complexity, we would expect a score of the inductive policy to occur between those for the optimal policy for the instance and the trivial policies. We observe, on average, that the scores of the VEJDE agents were statistically close to the scores of the MLP agents. In practice, due to the somewhat unstable optimization that deep RL constitutes, both the MLP and GNN may converge to local minima. This is our primary explanation as to why VEJDE policies scored higher than MLP policies on some domains. We could strengthen our evaluation by training five copies of MLP policies with some variation, such as different initialization values, to pair with the five VEJDE agents using different training sets. However, given the long training time required to train the 90 MLP policies, we opted against this. We hypothesize that our primary takeaway, that the scores of the inductive VEJDE policies is correlated with the transductive MLP policies, will still hold.

### 7.1.3 Prost

PROST receives the highest scores on most problems, showing its strong capability to search for optimal actions. However, the main drawback of PROST is that it is always dependent on an accurate generative model of the problem during test-time inference. This can be difficult to accommodate for use-cases where an accurate system model is not available, or the execution environment does not have the resources to perform continuous simulations. In contrast, the RL policies only need the simulator during training but do not perform any planning when choosing actions. The cost of the high scores with PROST is also paid with a much higher inference time, which we had to count in days, as opposed to minutes for the RL policies.

---

[17]The practical benefits of having to handle one agent per domain as opposed to ten is hard to quantify, but not insignificant.

Despite a strong overall performance, PROST performs little better than the do-nothing policy on certain problem instances, most notably from the domain *AcademicAdvising*. We are unsure if this is caused by PROST not being well-adapted for this particular problem, or the instances being defined such that doing nothing is indeed the optimal course of action. The higher average scores of the VEJDE agents suggests the latter may not be the case, however. The low scores of PROST on this domain is also reflected in the mimic policy, which are significantly worse on the problem compared to its RL counterpart.

## 7.2 Alternative State Representations

The maximum arity of facts across the problems we evaluate with is two. We could therefore represent the state for all included problems as a simple graph consisting of only nodes and edges, as Janisch (2024) or Garg et al. (2019). This is a format which is also known as Knowledge graphs (KGs), or RDF graphs, within certain research contexts (Ammanabrolu & Riedl, 2019). However, it is not common for works that handle KGs to incorporate numeric features, or object-specific features at all, which we require. Fey et al. (2024a) presents a graph representation directly influenced by relational databases, which extends simple graphs to include heterogeneous data. They group unary attributes of entities into fixed-sized vectors, analogous to the table columns in relational databases, forming a heterogeneous graph. The bipartite representation we use is more flexible in that entities can be represented with variably-sized sets of unary attributes. Being able to ignore missing or unseen attributes without padding is practical for problems where there are many possible object attributes, but only a small subset are present at a given time. We may, for instance, want to define many possible alerts for hosts in a network intrusion detection system, while only including the alerts that have actually been observed in the agent input.

In addition to changing the graph structure, there are also alternate methods for encoding the information in the graph, such as the graph attention method we describe in Appendix H. The conceptually simpler WL algorithm can be used instead of neural message-passing to encode the nodes and graph into latent states. However, for domains with continuous state variables the number of possible states becomes infinite, and we lose the ability to interpolate between latent states. Modifications can be made to the WL to allow for continuous representations (Chen & Thiébaux, 2024). We have also noted methods for belief propagation on factor graphs which incorporate neural networks, such as the one by Kuck et al. (2020), which could provide a different method for message-passing compared to the WL algorithm.

## 7.3 Limits to Generalization

Problems defined in RDDL arguably represent an ideal situation for structural generalization. The dynamics and reward are specified in a factored and lifted manner, meaning that we can map objects to their respective types and not lose instance-specific information. The dynamics are also constant across problem instances and the observations use the exact same predicates that are given to the agent, meaning that the value function and optimal policy can be defined using the language. In a realistic setting, there may be a disconnect between the data abstractions and the system they represent. This arguably forms a partially observable MDP, or a "context" MDP as described by Kirk et al. (2023) which each require more complex solutions methods than for the fully observable problems we consider in this evaluation.

# 8 Conclusion

We have developed and evaluated a framework that combines graph learning with model-free RL to find inductive policies for structured MDPs, which we call *Vejde*. The design allows policies to handle states which vary in both size and structure, as well as variable amounts of possible actions. We evaluated Vejde policies on eight problem domains defined with the Relational Dynamic Influence Diagram Language, training policies both by examples provided by the planner Prost, and with reinforcement learning. From our results, we found that the Vejde policies had, in average, a test performance on unseen instances not significantly different to MLP policies trained on each individual instance We also found that on some problem domains, policies trained with RL performed significantly worse than the corresponding policies trained with imitation learning. This leads us to a conclusion that different optimization procedures may be needed, such as improved exploration, to close this difference. Given that we as a society store much of our data using relational databases, we see a need to explore methods that allow us to apply neural networks to the data in the way it is stored. Previous work has explored the use of neural graph learning on databases for supervised learning, and in this work we show that a similar methodology can be used for reinforcement learning. Incorporating structure like relational databases provides in the design of the architecture facilitates structural generalization of agent policies, which we believe improves the real-world usability of reinforcement learning solutions.

### Acknowledgments

The authors would like to thank the authors of the pyRDDLGym library for making the problem domains, the MLP policy and Prost easily accessible for evaluation, and for responding to questions that arose during the work.

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

# 9 Appendices

## A Example State and Actions

Given the predicates $\{P, Q, Z, C\}$ and objects $\{x_1, y_1, y_2\}$, a particular state can be described as the set $\{P(x_1), Q(x_1, y_1), Z(y_1) = 5, Q(x_2, y_1), C\}$. Given the action symbols $\{A_1, A_2\}$, which can have different arities and type restrictions, a possible set of actions for the state may be $\{A_1(x_1), A_1(x_2), A_2(x_1)\}$. In the resulting bipartite graph, $U = \{x_1, y_1, x_2\}$ and $V = \{P(x_1), \ Q(x_1, y_1), \ Z(y_1) = 5, \ Q(x_2, y_1), \ C\}$.

## B Implementation Details

This appendix contains details which are less important in theory, but are relevant to the practical aspects of the implementation.

### B.1 Batching

Since each sample can vary in size, the common batching method of stacking vectors can not be used unless padding is added. We opt for the batching method used by graph learning code libraries such as PyTorch Geometric (Fey & Lenssen, 2019), where the adjacency matrix of the graph is stored in a COO matrix format. Thus, the node and adjacency vectors for each graph are concatenated in order to produce a single, albeit disconnected, graph.

### B.2 Action Masking

We calculate the probability of each action symbol for every object in the state, leading to the joint probability function including literals that are not possible according to the argument types of the action symbols. Under the assumption that taking invalid actions is equivalent to doing nothing, this leads to unnecessary exploration during training, as the agent has to explore a potentially large number of parameter combinations that will never be viable. We thus mask out parameter combinations that are incorrect according to the types of the action symbol. In practice, this means that the weights of invalid predicate-object combinations are assigned large negative values, so that the corresponding probabilities becomes zero.

### B.3 Code Library Design

VEJDE is designed to be domain-agnostic, and is built upon the Gymnasium interface (Towers et al., 2024). The heart of the implementation is a relational data model class, where predicate symbols, action symbols and object types of the particular problem domain are specified. This class is then used to shape the GNN and construct the graph representation of the state. We use a RDDL-specific instance of this class for our evaluation, which pulls the required information from a RDDL domain specification. In order to apply the library to a new problem domain, a relational data model for that domain has to be defined.

### B.4 Score Normalization

Returns are normalized according to the method used in IPPC 2014 (Vallati et al., 2015): For each instance, a lower bound return, $R_{low}$, is the maximum average return received from taking random actions or doing nothing. A maximum return, $R_{\max}$, for an instance is the highest return obtained among the evaluated methods. The score of an agent on a given instance is then $\max(R - R_{low}, 0)/(R_{\max} - R_{base})$, where $R$ is the average return over a given number of episodes. Thus, a score of 0 means that an agent performs worse or equal to acting at random or doing nothing, and 1 represents always having the highest score among the compared agents. We emphasize that a score of 1 does not imply that the agent follows the optimal policy of the MDP, which we do not have access to for comparison.

| Parameter Name | Value |
|---|---:|
| **Graph Neural Network** | |
| Embedding size | 16 |
| Activation function | tanh |
| Message passing layers | 4 |
| Critic prediction heads | 2 |
| Aggregation function | max |
| **Proximal Policy Optimization** | |
| Policy ratio clip factor | 0.2 |
| Entropy loss coefficient | 0.1/0.001/0.0001 |
| Critic loss coefficient | 1.0 |
| GAE $\lambda$ | 0.95 |
| Rollout steps | 1024 |
| Update epochs | 10 |
| **General Optimization** | |
| Optimizer | Amsgrad |
| Max. grad.2-norm | 1.0 |
| Total steps | 1 500 000 |
| Batch size | 16 |
| Number of parallel envs. | 16 |
| Discount factor | 0.99 |
| Learning rate | $10^{-3}/10^{-4}/10^{-5}$ |
| Exponential average $\alpha$ | 0.99 |

Table 6: Hyperparameters for reinforcement learning of Vejde policies. Learning rate and entropy coefficient was lowered every 500 000 steps.

## C    Hyperparameters

We used the same set of hyperparameters while training the GNN for all domains, which were chosen manually based on training time and returns observed in preliminary experiments on the training set. A rudimentary annealing scheme was used while training Vejde agents, where the learning rate was lowered by a factor of 10 for every 500 000 samples. The coefficient of the entropy term was also lowered after 500 000 samples, to guard against the policy converging to local minima early.

A source of difficulty in assigning a single set of hyperparameters was that the return values of the different domains, and even instances within the same domain, can have vastly different magnitudes. To improve the robustness of the optimization, we used two tricks from Dreamer (Hafner et al., 2025). The first is to scale $R_t$ and the value estimate $\tilde{v}$ by the *symlog* function, and the second is to scale the advantage $A_t$ by an exponential moving average of the return value range. The advantage $A_t$ was calculated using generalized advantage estimation (GAE) (Schulman et al., 2016), as is common in implementations of PPO.

Table 6 summarizes hyperparameters used in Vejde during experiments .

## D    Problem Selection & Extensions

In selecting the domains for evaluation, we excluded problems based on a set of filters, which we comment on in this appendix as stepping stones for extensions to the work.

### D.0.1    Continuous Actions

Since the policy is parameterized by neural networks, and PPO has previously used for continuous action selection, including actions with continuous values is theoretically straightforward but would require changing

the formulas for action probabilities and value estimations to account for the additional choice that the action value represents. Currently, the value for all actions is assumed to be a logical "True".

### D.0.2 Higher-Order Actions

We define higher-order actions as actions which involve more than one object. If we set the limit to at most binary actions, predicting the action is roughly equivalent to link prediction in the greater context of graph learning. Implementations of higher-order actions tend to require that tuples of objects are compared, which scales poorly if done naively (Ståhlberg et al., 2025; Ammanabrolu & Riedl, 2019; Morris et al., 2019). For binary actions, the number of argument combinations scales quadratically, trinary cubically and so on. Autoregressive sampling of action arguments, as done by Janisch (2024), is an alternative method for producing higher-order actions which scales better than evaluating every possible tuple, but requires a solution for representing partial actions to the decoding model.

### D.0.3 Concurrent Actions

The decentralized nature of MPNNs makes them suitable for implementing concurrent action selection, *i.e.* agents performing more than one action in a given timestep. Janisch (2024) demonstrated this with the *SysAdmin* domain, where the problem is changed to the agent selecting a subset of hosts rather than a single hosts. Having the agent select one action for each object, or a subset of objects, moves us closer to a cooperative multi-agent reinforcement learning formulation (Foerster et al., 2018), where we can view the graph as consisting of an arbitrary number of conditionally independent policies that should act in a cooperative manner.

### D.0.4 Partial Observability

Including problems with partial observability, or hidden *context* variables as defined by Kirk et al. (2023) would, in our opinion, improve the practical usability of VEJDE. Finding inductive solutions to problems with partial observability would likely require the addition of memory to the agent, or a temporal component to the database.

### D.0.5 Action Preconditions

This filter came about as a consequence of how `pyRDDLGym` handles action constraints. Certain domains in `rddlrepository` define logical constraints that limit what actions an agent is allowed to execute given the current state. If an agent executes an action which violates a constraint, the options in the simulator are to either crash, or put the simulator into an undefined state. A naive solution to this problem is to check if actions violate a constraint, and replace them with a default action, but not all domains define default actions. Another solution is to repeatedly sample the policy until a legal action is picked, but this requires a tighter coupling of the policy and environment than the Gymnasium interface typically defines.

### D.0.6 Nullary Domains

Though we did not formally include it in our results, we also tested VEJDE with the RDDL definition of the classic control problem CartPole with discrete actions. In this relational context CartPole becomes somewhat of an edge-case, as it is defined with only nullary predicates and actions. The policy decision is thus solely based on the aggregation of nullary predicates, described in 3.2. While recognizing that it constitutes a somewhat overcomplicated solution to the problem, we anecdotally found that VEJDE was able to find a satisfactory policy for the domain.

## D.1 Problems not Defined in RDDL

While we do not focus on problems not defined in RDDL in this paper, our implementation is designed to be used for other, yet relational, problem areas. We chose to use RDDL for our evaluation as it allows us to evaluate VEJDE on a varied selection of abstracted problem types without much additional implementation work. For the purpose of developing policies for automated defense, incident response has been simulated in

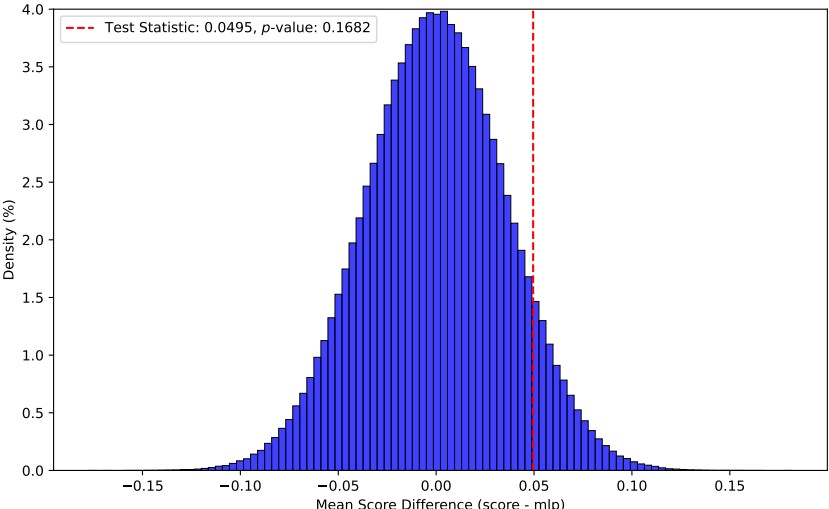

Figure 2: Null distribution from permutation test between average scores of VEJDE and MLP agent on all problems. $p$-value is calculated as $P(X > 0.0495) \cdot 2$.

a number of works, often as a two-agent game between an attacking and defending agent (Wolk et al., 2022). One of the more prominent examples of network incident response simulation is the CAGE set of problems, of which CAGE 4 is the most recent instance (Kiely et al., 2025). The problem state in both simulated and real-world incident response is often derived from log and system data, which tends to be relational by design and thus fits the relational reinforcement learning paradigm fairly well, as demonstrated by Thompson et al. (2024) for instance.

## E    Notes on Navigation

The goal of Navigation is for the agent to reach a given location, at which point a reward of 0 is given for each step. All IPPC problems uses 40 timesteps, meaning that if the agent reaches the end early the optimal action is for the agent to do nothing. This leads to a significant number of state-action tuples in the data collected from Prost consisting of Prost doing nothing. This skewed data distribution seems to impact mimic training negatively. With an extended amount of epochs over the training data, mimic performance improved significantly. Improved results was also obtained when all state-action tuples from after the agent has reached the goal were removed from the training data. This does not explain the RL agent's poor performance on Navigation, but it does answer the question of whether the architecture can encode a policy that does well on the problem.

## F    Plots from Permutation Test

Figures 2 and 3 show null distributions produced by sampling permutations of the pooled data and calculating the average score, with the observed average score marked with a red line. $10^6$ permutations were used. Null distributions from PROST comparisons are not included as the probability of the test statistic is 0.

## G    Box plots of scores per domain

Figure 4 shows box plots of average scores for each agent type for each domain.

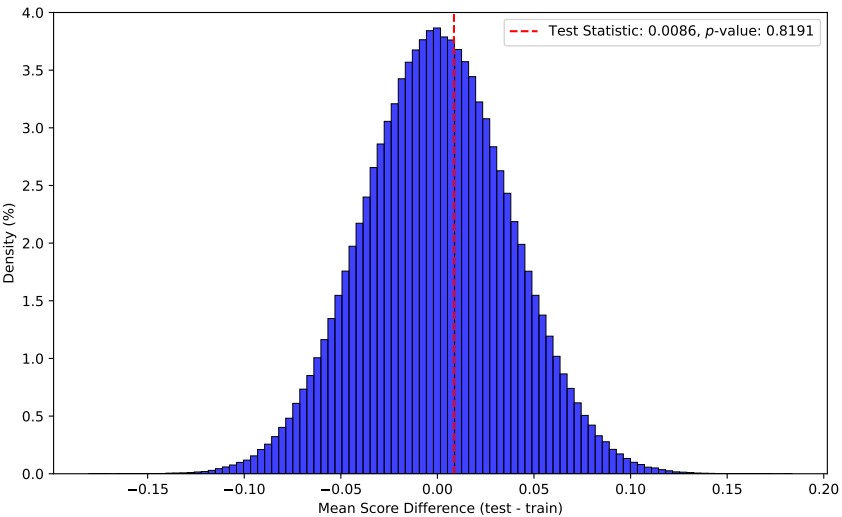

Figure 3: Null distribution from permutation test between average scores on test and train set problems for imitation learning agents. $p$-value is calculated as $P(X > 0.0086) \cdot 2$.

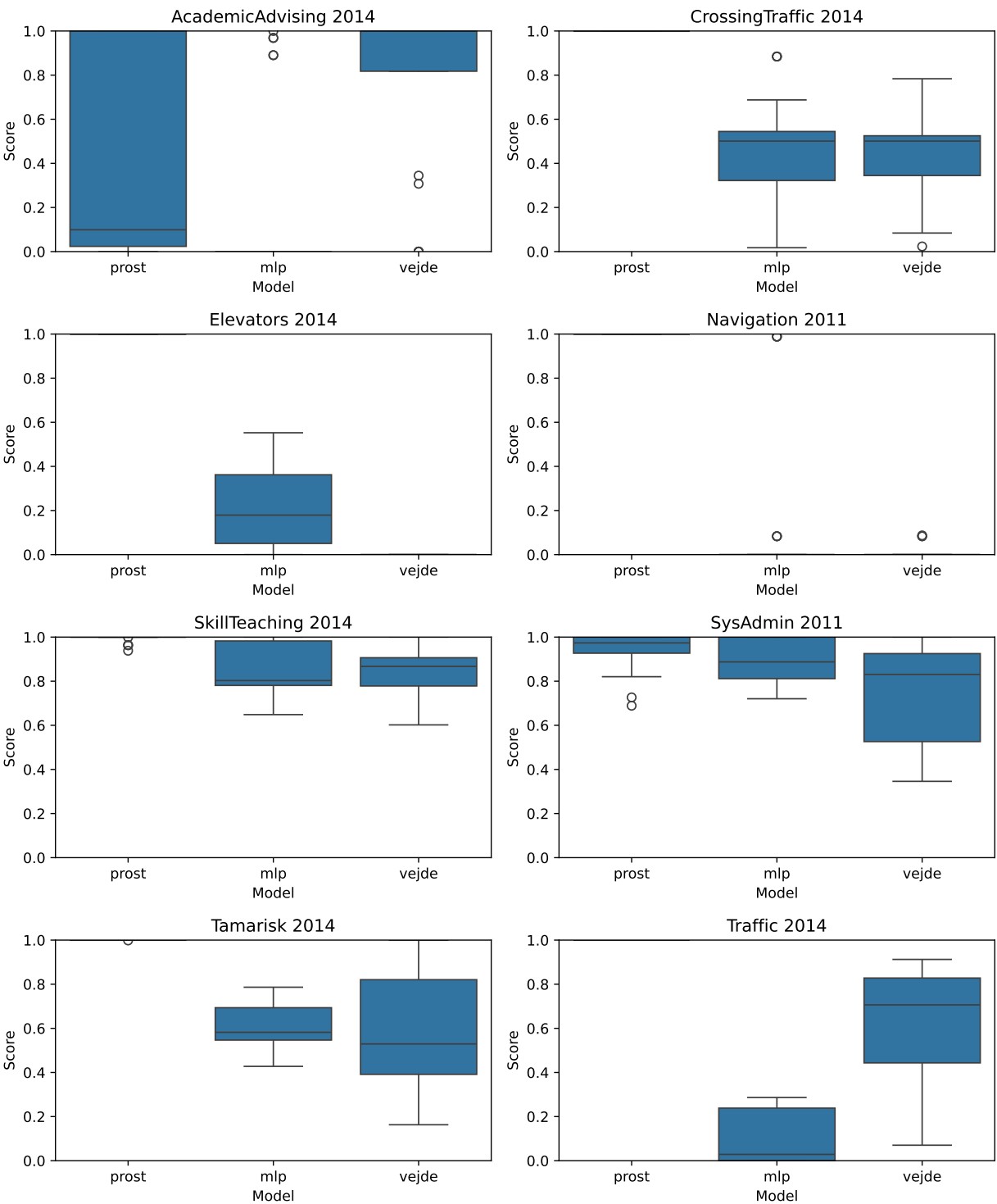

Figure 4: Box plots of normalized scores of Vejde agents, MLP agents and Prost for each domain.

# H Graph Attention

## H.1 Introduction

In addition to the message passing procedure described in Section 3.3, we also implemented and evaluated an encoder based on *graph attention* mechanisms. The primary difference between the local message-passing and attention-based graph representation methods is that the latter typically incorporates information from the entire graph at each step rather than the local node neighborhood. The attention function is technically message-passing over a fully connected graph, but for the purposes of this evaluation we will distinguish between graph attention and message-passing, with the latter referring to aggregations over node neighborhoods.

Dot-product attention does not incorporate the graph structure, meaning that locality information is lost (Ying et al., 2021). For instance, given the example shown in 1a, a policy using only dot-product attention can not distinguish between $x_1$ and $x_2$, assuming both are of the same type. This means that the naive application of a transformer in this context is bound to perform poorly in this context, which we also confirmed in preliminary experiments.

A remediation to the position invariance of attention is to explicitly include positional information with the input sequence. For sequential data, such as natural language, sinusoidal vectors derived from the element positions or relative distances have been used (Su et al., 2024). This approach fails for graphs, however, as graphs do not have a natural ordering. Instead, various properties that is derived from the adjacency matrix has been used, such as the degree, pairwise distances or graph Laplacian (Ying et al., 2021; Zhang et al., 2023).

## H.2 Implementation

Given that the state graph is bipartite, we chose to define the attention function

$$\alpha^{(i)}(X, Y) = \text{Softmax}((W_x^{(i)} X)(W_y^{(i)} Y)^T)$$

as a function of the two node sets, meaning that the resulting attention matrix consists of one weight for each pair in the cartesian product of the two sets. The attention function alone do not capture any graph structure, so in line with Zhang et al. (2023) we add the pairwise shortest-path distances between the objects and facts to the attention matrix. In line with previous work, distances are treated as indices that are mapped to learnable scalars. The embedded distance matrix is then multiplied element-wise with the attention matrix. The infinite distance, such as between disconnected nodes, is assigned a learnable scalar as well. The resulting, augmented, attention matrix is thus defined as

$$\eta^{(i)}(X, Y) = \alpha^{(i)}(X, Y) \odot \phi_D^{(i)}(D)$$

where $\odot$ denotes element-wise multiplication. Updates to the both node sets are done analogously to the message-passing method, but with $\eta$ replacing the adjacency matrix:

$$H^{(i+1)} = \phi_U^{(i)}(\eta^i(K^{(i)}, H^{(i)})K^{(i)} \parallel H^{(i)})$$
$$K^{(i+1)} = \phi_V^{(i)}(\eta^i(H^{(i+1)}, K^{(i)})H^{(i+1)} \parallel K^{(i)}) + K^{(i)}$$

## H.3 Experiments

We conduct the experiments with the graph attention policy with the same hyperparameters as for the message passing policy. The two architectures have roughly the same number of parameters ($\approx 12000$). The transformer theoretically requires less message-passes, as information is distributed across the entire graph in one step, but still benefits from additional layers. The imitation learning and reinforcement learning experiments was conducted in the way as described in Section 4.3.1. Due to the higher memory requirements

Table 7: Average normalized score per domain on test instances for PROST and graph attention policies trained with imitation learning.

| Domain | GA $\mu \pm \sigma$ | PROST $\mu \pm \sigma$ |
|---|---|---|
| SysAdmin | $0.71 \pm 0.41$ | $0.85 \pm 0.39$ |
| SkillTeaching | $0.60 \pm 0.22$ | $0.99 \pm 0.31$ |
| Tamarisk | $0.42 \pm 0.22$ | $1.00 \pm 0.31$ |
| CrossingTraffic | $0.31 \pm 0.42$ | $1.00 \pm 0.32$ |
| Traffic | $0.12 \pm 0.34$ | $1.00 \pm 0.15$ |
| Elevators | $0.10 \pm 0.75$ | $1.00 \pm 0.51$ |
| AcademicAdvising | $0.00 \pm 0.44$ | $0.80 \pm 0.66$ |
| Navigation | $0.00 \pm 0.00$ | $1.00 \pm 0.61$ |

of the graph transformer, we were forced to train the mimic with mini-batches rather than with the full dataset as with the message-passing neural network. The number of epochs over the data was kept the same, however.

## H.4 Results

We found that the graph attention policy produced produced significantly lower scores than the message-passing policy in both the imitation learning and reinforcement learning experiments. In addition to the lower scores, the training time of the attention-based policy was significantly higher for some domains, owing to the quadratic scaling of the attention function. The training times compared to the message passing policy are shown in Figure 5. The scores of the message passing agent and transformer agents for each domain are also shown side-by-side in Figure 6.

The full results of the imitation learning agent is shown in Table 7, and the RL agent in is shown in Table 8.

## H.5 Discussion

The somewhat poor results of the graph attention method is consistent with results we have observed in previous work, where the method that incorporated graph-wide information had worse generalization properties than the entirely local method Nyberg & Johnson (2024). The message passing policy is not entirely localized either, however, as the first action is made based on information from the entire graph, but the message passing is local.

The graph attention implementation adds two major compute costs. One is in calculating pairwise distances, which we do for each step[18]. The second additional compute cost is memory. Due to the dot-product in the aforementioned dot-product attention, the resulting attention matrix will scale quadratically with the number of nodes in the graph.

We recognize that the graph attention method we evaluated is a somewhat basic implementation. Additional steps could be added, such as a self-attention step, and different graph metrics can be added to the attention matrix, but we opted for a simple initial approach that still incorporates the main components of Transformer-esque models.

---

[18]With caching, the worst case is every unique state. However, our attempts to cache the distances for each state led to system memory running out for problems with large state spaces.

Table 8: Average normalized scores on test instances for graph attention agents trained using RL and PROST, per problem domain.

| Domain | GA $\mu \pm \sigma$ | MLP $\mu \pm \sigma$ | PROST $\mu \pm \sigma$ |
|---|---|---|---|
| SkillTeaching | $0.58 \pm 0.34$ | $0.83 \pm 0.23$ | $0.99 \pm 0.31$ |
| Tamarisk | $0.32 \pm 0.18$ | $0.61 \pm 0.18$ | $1.00 \pm 0.31$ |
| CrossingTraffic | $0.27 \pm 0.47$ | $0.46 \pm 0.50$ | $1.00 \pm 0.32$ |
| SysAdmin | $0.26 \pm 0.37$ | $0.93 \pm 0.34$ | $0.95 \pm 0.47$ |
| Traffic | $0.14 \pm 0.39$ | $0.12 \pm 0.27$ | $1.00 \pm 0.15$ |
| Navigation | $0.01 \pm 0.25$ | $0.12 \pm 0.29$ | $1.00 \pm 0.61$ |
| AcademicAdvising | $0.00 \pm 0.39$ | $0.29 \pm 0.09$ | $0.80 \pm 0.66$ |
| Elevators | $0.00 \pm 0.57$ | $0.20 \pm 0.52$ | $1.00 \pm 0.51$ |

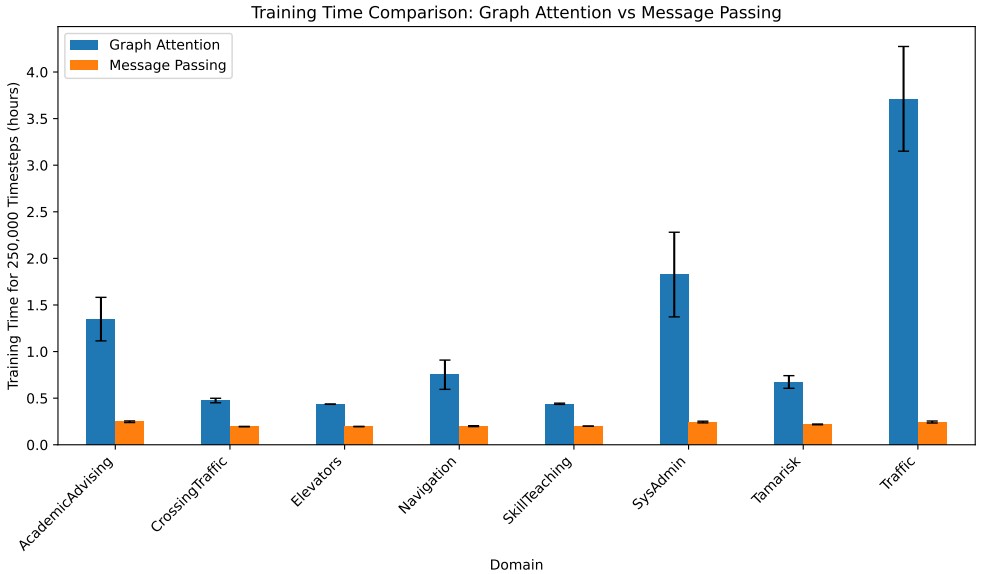

Figure 5: A bar chart showing training times in hours for message passing agents and graph attention agents on each domain. Times were averaged over five runs.

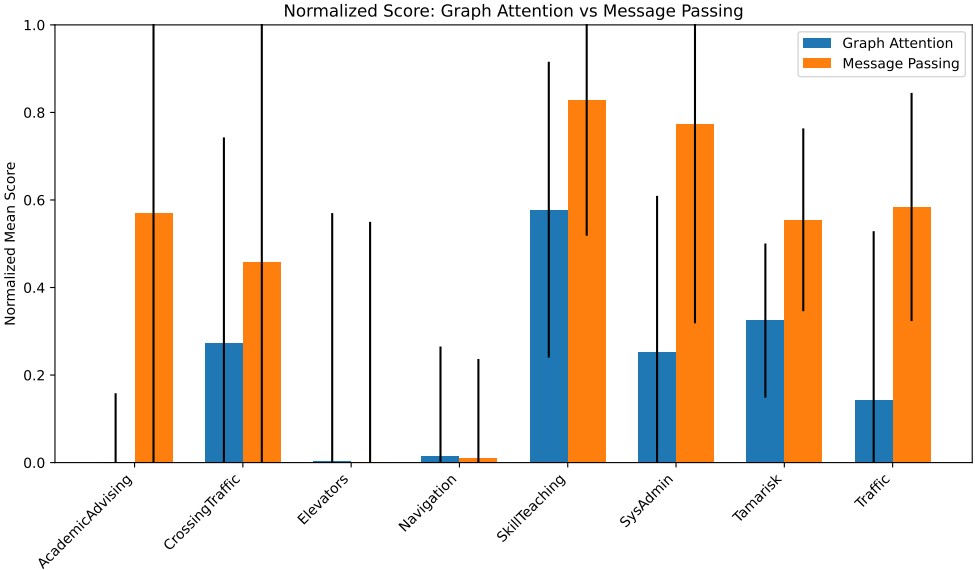

Figure 6: A bar chart showing normalized scores for message passing agents and graph attention agents on each domain. Scores were averaged over five runs.

