# OpenReview forum: "Vejde: A Framework for Inductive Deep Reinforcement Learning Based on Factor Graph Color Refinement"
_TMLR — Accepted by TMLR_

### Review · Reviewer_LTcH · 2025-09-15

**Summary Of Contributions:**

This paper proposes a novel graph-based RL to learn inductive policies for structured MDPs. The states of the MDP are represented as the data bases of fact and then encoded as embeddings using the neural message passing. A policy parameterized by neural networks outputs the probability of the actions over the states. In practical implementation, some techniques including batching and action chunking are used. Experimental evaluated are conducted on 8 problem domains from RDDL. The results show that the learned policy achieves good generalization, without a significant loss in performance.

**Audience:**

Yes

**Audience Explanation:**

Readers interested in RL may find this paper of interest.

**Claims And Evidence:**

Yes

**Claims Explanation:**

The statements are supported by empirical results.

**Requested Changes:**

The empirical results show that VEJDE achieves worse average performance than the MLP policies on 5 of 9 problem domains. These results are fine for me, since the MLP policies are trained on each problem instance. However, I notice that the paper does not show the comparison of VEJDE to other inductive policies. Adding a comparison to similar inductive policies is helpful to show the benefits of the proposed method, particularly when its performance does not surpass the MLP policies. Although the authors explain that they cannot find a single method that covers the set of problem domains, providing comparisons on a subset of problem domains is still helpful.

---

> ### Author Response · Authors · 2025-11-10
> **Incorporating previous works is difficult while preserving their independence**
>
> Thank you for your review. We strongly agree that comparing Vejde to another method for inductive policy representation would be both helpful and useful (since both the MLP and Prost baselines are transductive), but we are not aware of similar methods from previous works that we can incorporate in our evaluation without significant additional implementation and design choices. If you have a concrete suggestion of a method you would like included, we would be gladly incorporate it in the evaluation. At the suggestion of Reviewer gZxp, we have implemented an alternative embedding method based on graph attention, which at least changes the graph embedding methodology while keeping the rest of the pipeline the same. Descriptions and evaluations of this method has been added to the revised paper.
>
> The works mentioned in our related works fall into two primary categories: those that use similar representations but for planning, and those that use similar representations with reinforcement learning. The works that use reinforcement learning are fairly specialized for the problems they selected in their works. Porting them to the problems that Vejde under-performs at would require a significant amount of original implementation work, and perhaps more importantly, design decisions not made by the original authors.
>
> I think the main point that is different to, and representative of, both SR-DRL and TrapsNet is that they represent the state as a simple graph, rather than a bipartite graph as Vejde does. This typically means that object properties (i.e. unary relations) are stored 'inside' the nodes and encoded, e.g. passed through a neural network layer, before doing message passing between nodes.
>
> Implicit in this approach is that there is an explicit separation of unary and binary predicates in the message passing architecture, which is a distinction that Vejde does not make. I think the further generalization to this is to separate ground values by their arity, which is something [Ståhlberg et. al.](https://arxiv.org/abs/2403.11734) has done in their work with deep automated planning.

---

### Review · Reviewer_gZxp · 2025-09-30

**Summary Of Contributions:**

The paper proposes Vejde, a framework that works on MDP of relational databases of facts about entities, converts each state to bipartite graph, use of GNN architecture and trained with PPO objective. The key idea is that factored representation of both states and actions allows Vejde agents to handle problems of varying size and structure. The evaluation results are on eight problem domains defined in
RDDL, with ten problem instances each. When initialized via imitation learning, Vejde achieves performance comparable to an online planning algorithm. With reinforcement learning, Vejde’s GNN-based agents outperform MLP baselines, specifically in the AcademicAdvising and Traffic domains.

**Audience:**

No

**Audience Explanation:**

The topic might be of interest to TMLR's audience. But people will appreciate the findings of the paper if it is revised from its current form. Here are the three reasons:

1. The relational scope is limited. By excluding higher-arity predicates and simplifying domains, the setup no longer reflects the complexity of real relational databases, making the tasks less compelling for the broader ML community.

2. The architectural choice is focused on using GNNs with color refinement and compared to MLPs. But GNN with bipartite graph on each state seems less flexible than Transformers, which could more naturally model subject–predicate interactions and scale beyond the small RDDL datasets used.

3. While the paper covers exhaustive details, it is overly long. The draft could be condensed to about 12 pages, with standard background and detailed implementation moved to appendices, improving clarity and impact.

**Broader Impact Concerns:**

broader impact is not discussed in the paper.

**Claims And Evidence:**

Yes

**Claims Explanation:**

The experiments are well structured for supervised and RL training results provided over five randomly sampled train/test splits and
repeated experiments for each split. The online planning algorithm PROST provides a good upper bound to compare with and p_null value is discussed puts the results into perspective.

**Requested Changes:**

1. Clarify why the problem is still interesting and novel despite the simplification. It seems that the key aspects of RDDL has been filtered out in the current evaluation.
1. Include transformers as a baseline, at least for the imitation learning experiments. What is needed to cover the clear gaps in Navigation (0.33 vs 0.99) and Traffic (0.73 vs 1.00) environments?
1. Make the paper concise in 12 pages and move extra details to Appendix.

---

> ### Author Response · Authors · 2025-11-10
>
> Thank you for your review. I will try to respond to your questions piecemeal below.
>
> > The architectural choice is focused on using GNNs with color refinement and compared to MLPs. But GNN with bipartite graph on each state seems less flexible than Transformers, which could more naturally model subject–predicate interactions and scale beyond the small RDDL datasets used.
>
> Transformers are [well documented](https://proceedings.neurips.cc/paper/2021/file/f1c1592588411002af340cbaedd6fc33-Paper.pdf) to perform poorly on tasks involving graph/relational data. The main reason for this is that dot-product attention does not encode position or locality.
>
> This means that if we naively apply dot-product attention to the bipartite graph, the policy using the resulting embedding will, for instance, not be able to tell which properties belongs to which objects. Take the SysAdmin problem as an example: If a single machine shut off, every other object would be updated with that information, and thus look the same from the perspective of the policy function.
>
>  In sequence processing the position invariance of attention is usually adressed by adding different position encodings, such as with sinusoids. However, graphs do not have such a natural ordering, so different graph metrics tend to be used instead.
>
> Taking a few cues from [2301.09505](https://arxiv.org/pdf/2301.09505), We have now implemented a version of the "Graphformer", which supplements the attention matrix with the pairwise distances between nodes. The main benefit of graph attention is that the entire graph is encoded in a single pass, alleviating the need to set a number of message passes.
>
> We have add an evaluation of the graph attention method and a comparison to the message passing method to the revised paper.
>
> ---
>
> > Clarify why the problem is still interesting and novel despite the simplification. It seems that the key aspects of RDDL has been filtered out in the current evaluation.
>
> I am a bit unsure what "key aspects of RDDL" you are referring to. To us, the key aspect of RDDL is the lifted problem specification that is grounded in different problem instances. A Vejde agent is defined using the lifted specification, which gives a bound of the agent's generalization capability. The elements which we have simplified are:
>
> - Multiple actions per time step. The RDDL problems we simplified has the agent select one action per object, rather than one action per time step (or graph as one can also see it). This changes the problems, but I would not say it invariably makes the problems simpler since it adds an aspect of prioritization to the tasks (since the agent can not control all entities at the same time).
> - We ignored the problems from IPPC with higher-order **actions**. This has the side-effect of also excluding problems where fluents have arities higher than two, which is the nature of the problem set considered.
>
> It is true that we do not demonstrate the ability to encode higher-order predicates. However, this mainly comes from not implementing higher-order predicates in the _output_. As we note in the discussion, higher-order outputs is a relevant addition, but incur a potentially high performance cost and requires some additional design choices. I do agree on their importance though, at least when it comes to binary actions.
>
> ---
>
> > The relational scope is limited. By excluding higher-arity predicates and simplifying domains, the setup no longer reflects the complexity of real relational databases, making the tasks less compelling for the broader ML community.
>
> I am again a bit unsure what the "complexity of real relational databases" relate to here. Perhaps you can elaborate on what you feel is missing? I agree that most relational databases have relations that involve more than two entities. Vejde is able to handle such higher-order predicates in the _input_ (i.e. the observation space), even if it is not able to do so in the _output_ (the action space). It is true, however, that none of the problem domains considered have higher-order predicates in the _input_ space. This comes as an unfortunate side-effect of us filtering out domains with higher-order _actions_, which I have discussed above.
>
> > scale beyond the small RDDL datasets used.
>
> While not the same size as [the largest graph datasets](https://pytorch-geometric.readthedocs.io/en/latest/cheatsheet/data_cheatsheet.html), a batch of 1024 steps does contain a fair amount of data.
>
> Take for example one of the larger instances, SysAdmin 10, which models 50 hosts.
>
> This means that in each observation/step there are:
> - 50 entity nodes
> - (at worst) 50 unary properties.
> - ~142 relations.
> - 1 global node
>
> We use 1024 steps in a batch so that amounts to 256 000 nodes stored in a single disconnected graph that is embedded with the message passing architecture.

---

> ### Author Response · Authors · 2025-11-10
> **Notes on Navigation**
>
> > What is needed to cover the clear gaps in Navigation (0.33 vs 0.99) and Traffic (0.73 vs 1.00) environments?
>
> For Navigation at least, one of the main reasons for the mimic performing poorly seems to be poor data distribution. Due to the way the end condition is set up in the RDDL domain, there is a large number of state-action pairs where Prost does nothing due to having already found the goal. This is because the episode does not end when the agent reaches the goal, but rather after 40 time-steps (all the IPPC domains work like this). I have now done a short experiment where I manually removed all the state-action pairs with NOP actions and it significantly increased the convergence speed and performance of the imitation learning agent. We have added this information to the revised version of the article.
>
> Poor data distribution does not explain the poor performance of the Vejde agent in the RL experiment, however, which I believe is related to reward sparseness, similar to the Elevator domain. Navigation is also:
>
> - Highly reliant on global aggregation of the state since all actions are nullary.
> - The graph resulting from the state (which is essentially a grid) might make message passing more unstable. A bit of instability can be fine in the supervised situation, but might be too fragile for PPO to work.

---

### Review · Reviewer_zxeU · 2025-11-06

**Summary Of Contributions:**

This paper presents Vejde, a framework combining graph neural networks (GNNs) and reinforcement learning (RL) to produce inductive policies that generalize across problem instances within a domain. Vejde represents MDP states as bipartite factor graphs derived from relational databases, uses message-passing neural networks (MPNNs) to generate latent representations, and applies PPO for training.
The authors evaluate Vejde on eight RDDL domains, using both imitation learning (from Prost) and RL, comparing results against instance-specific MLP agents and the Prost planner. Results show that Vejde generalizes to unseen instances with only minor performance loss and achieves scores close to instance-specific MLPs.

**Audience:**

Yes

**Audience Explanation:**

The topic is relevant, the proposed architecture is coherent, and the evaluation spans a comparatively large portion of RDDL benchmarks for relational RL and I think TMLR's audience can be interested in knowing the findings.

**Broader Impact Concerns:**

No broader impact concern

**Claims And Evidence:**

Yes

**Claims Explanation:**

The paper articulates the problem of RL generalization to relational domains well and situates it convincingly within relational and neuro-symbolic RL literature. The factor graph and color refinement formulation are well-grounded in prior work. The mapping from logic-based state descriptions to GNN representations is well-motivated.

**Requested Changes:**

The goal of the paper is important and aligns with a renewed interest in structured RL and neuro-symbolic generalization. The combination of a bipartite fact–object factor graph with MPNN message passing is straightforward, interpretable, and domain-agnostic. The framework should be re-usable in other relational RL settings. From what I know about previous literature, testing across 8 RDDL domain is substantially broader than what several prior relational RL works present. However I feel there are some issues. While MLPs and Prost are strong baselines, the lack of comparison to other inductive or graph-based RL methods is a missed opportunity. The authors acknowledge this but could better contextualize their results relative to prior work. Furthermore, the paper would benefit from an ablation study (e.g., the effect of message-passing layers, aggregation functions, or the bipartite graph structure) to justify design choices and sensitivity to hyperparameters (e.g., embedding size, learning rate schedules) is not discussed, which could affect reproducibility. Several domains show near-zero RL performance  This raises concerns: are these failures due to PPO instability, representational limits of the architecture, or training hyperparameters? why does imitation succeed in Elevators but PPO fails? The paper acknowledges this but provides no diagnostic analysis.
The paper has an ambitious goal and a clear design to model the problem it want to solve, but I feel it is still incomplete and with many directions of improvement. However, my confidence in the assessment is not very high as my expertise is more concerned with the theoretical nature of RL, so I may have missed the more recent developments in the field and I may not be aware of some of the most common and best practices of the literature, so I'm very open to change my mind if the authors provide convincing explanations.

---

> ### Author Response · Authors · 2025-11-10
>
> Thank you for your review. We will try to respond to your questions piecemeal below.
>
>  >While MLPs and Prost are strong baselines, the lack of comparison to other inductive or graph-based RL methods is a missed opportunity.
>
> > The authors acknowledge this but could better contextualize their results relative to prior work.
>
> We strongly agree that additional inductive methods would be good. However, the related work we are aware of is either
> + Not applicable to the entire problem range. This tends to imply a certain amount of design is needed to adapt them to the entire range.
> + Specific to planning. This tends to imply a certain amount of effort is needed to adapt it to the RL context.
>
> Both categories requires a not insignificant amount of implementation work, and perhaps more crucially, design decisions that fall on us to make, describe and potentially evaluate. At the suggestion of Reviewer gZxp, we have implemented an alternative graph embedding method based on graph attention, which at least changes the message passing method while keeping the rest of the pipeline the same. We have added the evaluation of this method to the revised paper.
>
> ---
>
> > are these failures due to PPO instability, representational limits of the architecture, or training hyperparameters?
>
> Through the imitation learning (IL) experiment, we show that, for most domains, there is a set of weights that can be assigned to the message passing neural network that allows it to perform almost as well as Prost. Since the only changing factor between the IL and RL experiment is the choice of learning algorithm, I would argue that this singles out PPO as the point of failure.
>
> As to _why_ PPO fails, we do not have a definitive answer. Our current theory as to why PPO fails with Elevators is because the problem has a relatively sparse and/or harsh reward function, with a heavy penalty for taking the wrong action. There is a local reward maximum one can arrive at by doing nothing, so some effort might be needed to avoid it.
>
> An optimization procedure that encourages more dynamic exploration, such as one with [Intrinsic Curiosity](https://arxiv.org/pdf/1705.05363) might fare better than vanilla PPO. However, Intrinsic Curiosity requires next-state prediction, which in this graph context requires some non-trivial design decisions and implementation work (mainly stemming from the fact that the graphs can change size between states).
>
> ---
>
> > Furthermore, the paper would benefit from an ablation study (e.g., the effect of message-passing layers, aggregation functions, or the bipartite graph structure) to justify design choices and sensitivity to hyperparameters (e.g., embedding size, learning rate schedules) is not discussed, which could affect reproducibility.
>
> There are roughly two categories of hyperparameters involved, those for the GNN and those for PPO. Both of these could in theory be studied further.
> The imitation learning experiment can be regarded as an ablation study when it comes to the RL component in its entirety, since it exchanges the learning algorithm while maintaining the GNN hyperparameters. The choice of message passing layers is a significant one, as it may strictly limit the expressiveness of the policy if set too low, but as we see from the IL experiment it should not be an issue for Elevator, for instance.  As mentioned above, we have included a test of an attention-based encoding method. It removes the message passing layers as a parameter but it also adds some other risks (for instance the choice of graph distance metric).
>
> Most of the PPO hyperparameter values we use are the defaults one would expect from a framework like SB3, as this is what I would suspect the average person would pick when training on a new domain. The remainder of our hyperparameter choices were made somewhat conservatively to avoid variability caused by the varying domains and the other hyperparameters. This includes:
>
> - A high number of iterations. 1.5 Million total is more than enough for most of the problems, and most agents reach stable policies and value estimates.
> - Excessive exploration for the first 500000 iterations (none of the policies remain at high entropy when the entropy parameter is lowered).
> - The DreamerV3 reward scaling methods. Both of these make training more stable at the expense of also making it slower in terms of required updates, since the reward signal is dampened and passed through an exponential average (also relates to the point above) .
> - Repeat experiments with different data/problem distributions in the training data.
>
> We are fairly sure that there is a set of hyperparameters that would let us train faster with less iterations, or lower the standard deviation of the results, but we consider the ones we use stable enough for the purpose of the evaluation.

---

> > ### Comment · Reviewer_zxeU · 2025-11-20
> >
> > Thanks for the positive approach to the suggestions, the changes made and the explanations. Now it's both cleaner the exposition and clearer the scope and why some choices were made. I will definitely take it in account in the final recommendation

---

### Decision · Action_Editor_RnJu · 2025-12-19

**Recommendation:** Accept as is

**Audience:**

Yes

**Audience Explanation:**

Of potential interest to many RL researchers.

**Claims And Evidence:**

Yes

**Claims Explanation:**

Appropriate experiments to back up the claims. Always possible to add more (e.g., more ablations, more hyperparameter testing), but current setup is reasonable undoubtedly sufficient to be of interest, and there is no overclaiming.